# Effects of dysmenorrhea on work productivity and quality of life in Japanese women: A large-scale web-based cross-sectional study

Maika Nariai[1], Osamu Wada-Hiraike[1]*, Eri Maeda[2], Masayo Matsuzaki[3], Mayuyo Mori-Uchino[4], Maho Furukawa[1], Yuki Enomoto[1], Hiromi Ga[1], Risa Takai[1], Miyuki Harada[1], Yutaka Osuga[5], Yasushi Hirota[1]

1 Department of Obstetrics and Gynecology, Graduate School of Medicine, The University of Tokyo, Tokyo, Japan, 2 Department of Public Health, Faculty of Medicine, Hokkaido University, Sapporo, Japan, 3 Department of Reproductive Health Nursing, Graduate School of Health Care Sciences, Institute of Science Tokyo, Tokyo, Japan, 4 Department of Radiology, Tokyo Metropolitan Cancer and Infectious Diseases Center Komagome Hospital, Tokyo, Japan, 5 Department of Obstetrics and Gynecology, Teikyo University School of Medicine, Tokyo, Japan

* osamuwh-tky@umin.ac.jp

## Abstract

### Research question

How significant is the impact of dysmenorrhea on work productivity and quality of life (QoL) among Japanese women?.

### Methods

This large-scale cross-sectional study was conducted using a web-based self-report questionnaire administered via a smartphone application (LunaLuna). The Work Productivity and Activity Impairment Questionnaire: General Health (WPAI-GH) and the SF-36v2 health survey were used to assess work productivity and health-related QoL, respectively. Work productivity and health-related QoL were compared between the severe and non-severe groups as defined by the dysmenorrhea score, which assesses the severity of menstrual pain, its impact on daily activities, and medication use. A multiple regression analysis was performed to estimate the magnitude of the effect of dysmenorrhea on work productivity and health-related QoL after adjusting for confounding factors.

### Results

In total, 2,555 women were included in the analysis. Among them, 2064 women completed the dysmenorrhea score questionnaire, and 902 (43.7%) participants with a dysmenorrhea score of ≥3 were categorized as the severe group. The median overall work impairment was 16% higher in the severe dysmenorrhea group than in the non-severe dysmenorrhea group. In the multiple regression analysis, despite

**Data availability statement:** The dataset used in this study is summarized and anonymized. It was submitted as the supplemental data and the data is available to the scientists upon reasonable request. All relevant data are within the paper and its Supporting Information files.

**Funding:** This study was supported by the Ministry of Health, Labour and Welfare (24FB1001: OWH, No number: YO).

**Competing interests:** The authors declare no possible conflicts of interest to disclose.

adjusting for confounding factors, the severity of dysmenorrhea remained a significant predictor of impaired absenteeism; presenteeism; overall work impairment; activity impairment; and reduced physical, mental, and social QoL. Additionally, low annual household income (<5,000,000 yen) was a significant predictor of reduced work productivity and lower mental and social QoL.

## Conclusions

The severity of dysmenorrhea significantly affects the work productivity and quality of life of Japanese women. A dysmenorrhea score of ≥3 as a threshold for treatment eligibility may be a useful and potentially valid criterion. The application of a dysmenorrhea score may facilitate the screening of dysmenorrhea in clinical practice.

### Introduction

Dysmenorrhea, defined as intense abdominal pain occurring immediately before and during menstruation, has a prevalence of 67%–90% [1]. Dysmenorrhea is common in women of reproductive age [2,3]. This age group corresponds to the prime working years, and the overall impact of menstrual symptoms on work productivity is substantial. The annual economic burden of menstrual symptoms for Japanese women is estimated at USD 8.6 billion [4], indicating the necessity for effective interventions.

Previous studies have reported that dysmenorrhea negatively impacts work productivity [5–7]. Additionally, its impact on quality of life (QoL) has been investigated in previous studies [8–10]. However, many previous studies have defined dysmenorrhea based on the self-reported presence of pain during menstruation or by using pain severity assessments, such as the Visual Analog Scale (VAS) or Numerical Rating Scale (NRS), through interviews [1], making it difficult to compare the results between different studies. Moreover, many Japanese women reported perceiving pain during menstruation as "natural," and 46.8% cited "not feeling the need" as their reason for not seeking medical care [11]. Therefore, self-reported surveys of Japanese women may have underestimated the true prevalence of dysmenorrhea. Moreover, 94.8% Japanese women reported using analgesics for menstrual pain [11], suggesting that their self-assessment of pain severity may have been underestimated by the regular use of analgesics.

The dysmenorrhea score [12], calculated as the sum of the pain and drug scores, can assess the severity of pain that individuals may not be consciously aware of, enabling a more appropriate evaluation of the disease. In previous studies, the dysmenorrhea score has been widely used in clinical trials in Japan for evaluating treatment efficacy [12–14]. However, few studies have assessed the validity of using dysmenorrhea scores to identify populations in untreated groups that require therapeutic intervention. Therefore, we aimed to analyze the impact of dysmenorrhea on work productivity and QoL using the dysmenorrhea score as well as the validity of using this score to assess the severity and need for therapeutic intervention in untreated populations.

## Materials and methods

### Data collection

Participants were recruited via the smartphone application LunaLuna (https://www.mti.co.jp/eng/?page_id=2755), which is an application for recording the menstrual cycle, managing menstrual schedule, predicting ovulation date, and tracking pill use, and is freely available on Android and iOS platforms [15]. Individuals who provided informed consent to participate in the study were administered two separate questionnaires. The first questionnaire assessed the dysmenorrhea score [12], while the second collected data on participants' background factors and the Work Productivity and Activity Impairment Questionnaire: General Health (WPAI-GH) and SF-36v2 health survey responses. Questionnaires were distributed in two separate sessions from June 2022 to August 2022.

The study protocol was approved by the Research Ethics Committee of the Graduate School of Medicine, University of Tokyo (ethics approval 2020374NI; approved on March 21, 2021). This study was performed in accordance with the principles of the Declaration of Helsinki. MTI Ltd. anonymized and transferred the data to the Graduate School of Medicine at the University of Tokyo. Inclusion in the study required participants to acknowledge the in-app notifications describing the research outline and data usage policy, followed by the selection of the Agreement button, thereby providing their informed consent electronically. The data used in this study were accessed for research purposes from February 2, 2025 to June 16, 2025. The authors did not have access to information that could identify individual participants during or after data collection; all data were fully anonymized prior to analysis.

### Measurement of the severity of dysmenorrhea

The dysmenorrhea score [12] is a composite measure used to assess the severity of menstrual pain and its impact on daily activities and medication use. The total dysmenorrhea score is calculated as the sum of pain and drug scores.

Pain score ranges from 0 to 3 points:

- 0 = No pain

- 1 = Mild pain (pain is present but does not interfere with daily life)

- 2 = Moderate pain (pain causes some interference in activities of daily life, but not enough to require rest)

- 3 = Severe pain (daily life is difficult or rest is required due to pain)

And the drug score ranges from 0 to 3 points:

- 0 = No use

- 1 = Once per menstrual cycle

- 2 = Twice per menstrual cycle

- 3 = Three or more times per menstrual cycle

The total score ranges from 0 to 6, and participants with a total score of ≥3 were categorized in the severe group.

### Assessment of work productivity

WPAI-GH [16] is a self-reported instrument that measures the impact of health problems on work productivity and daily activities over the past 7 days. It comprises four domains: absenteeism, presenteeism, overall work impairment, and activity impairment. Each domain is scored from 0% (no impairment) to 100% (complete impairment). Absenteeism is the percentage of work time missed owing to health issues. Presenteeism is the percentage of work impairment due to health issues. Overall work impairment was calculated using the following formula: Absenteeism + [(1-absenteeism)

× presenteeism]. Activity impairment is the percentage of impairment in daily activities outside of work owing to health issues.

## Assessment of health-related QoL

SF-36v2 [17,18] is a widely used self-administered questionnaire designed to measure general health-related (HR)QoL. The results consist of eight subscales [Physical Functioning (PF), Role Physical (RP), Bodily Pain (BP), General Health (GH), Vitality (VT), Social Functioning (SF), Role Emotional (RE), and Mental Health (MH)] and three summary scores [Physical Component Summary (PCS), Mental Component Summary (MCS), and role/social component summary (RCS)] [19]. Norm-based scores were used for evaluation; that is, each SF-36 domain and component summary score is standardized against the general Japanese population to have a mean of 50 and a standard deviation of 10.

## Assessment of cofounding factors

Many factors are associated with the severity of dysmenorrhea, the type of symptoms, work productivity, and HRQoL. Age, BMI, and comorbidities are known to be associated with the severity of dysmenorrhea [20]. In addition, psychological distress and dysmenorrhea are correlated [21], and the severity of PMS symptoms is related to the severity of dysmenorrhea [20,21]. Moreover, income and area of residence have been found to be correlated with QoL [22,23]. Therefore, to adjust for confounding factors, the following variables were collected using a questionnaire and were included in the model:

- Age ranged from 20 to 49 years and was included in the model as a continuous variable.

- BMI: Calculated from height and weight reported in the questionnaire

- Gravidity

- Area of residence: Based on the Japanese Resident Register as of 2024, prefectures ranked within the top 10 in population were defined as "urban areas," while all other prefectures were defined as "rural areas."

- Annual household income: <5,000,000 yen in the low-income group; between 5,000,000 and 10,000,000 yen in the middle-income group; and >10,000,000 yen in the high-income group.

- Comorbidities: Participants indicated whether they had no comorbidities or had any of the following diseases: hypertension (HT), dyslipidemia (DL), diabetes mellitus (DM), asthma, allergy, liver disease, heart disease, tuberculosis, autoimmune disease, anemia, stroke, cancer, and other comorbidities. Comorbidities with a low prevalence were excluded from the regression analysis, and HT, DL, DM, asthma, allergy, heart disease, and anemia were included in the model.

## Statistical analysis

The two surveys were linked by ID, and if there were duplicate IDs, only the record with the later timestamp was retained. Participants were excluded if they had not yet experienced menarche, were postmenopausal, or were already receiving hormonal therapy, such as low-dose oral contraceptives or intrauterine devices. The WAPI-GH and SF-36v2 values were compared between the severe and normal groups, as defined by the dysmenorrhea score. To compare background factors, continuous variables were analyzed using the Wilcoxon rank-sum test, whereas categorical variables were assessed using Fisher's exact test. For the multivariate analysis, a multiple regression model was constructed with absenteeism, presenteeism, overall work impairment, activity impairment, PCS, MCS, and RCS as dependent variables. Spearman's correlation coefficient was used for correlation analysis. R version 4.3.3 [24] and RStudio 2024.12.0 [25] were used for statistical analysis, and a p-value <0.05 was considered statistically significant.

## Results

### Descriptive statistics

In total, 2,974 and 3,677 women answered the first and second questionnaires, respectively. The total number of merged IDs was 3,723. After excluding women who were not menstruating and those who were already receiving hormonal therapy, data from 2,555 women were used in the final analysis.

Among the participants who answered the WPAI-GH questionnaire and whose absenteeism data were available, 366/2,096 (17.5%) (95% confidence interval [CI]: 15.8% to 19.1%) experienced absenteeism due to health reasons. Among participants whose presenteeism data were available, 1,311/2,040 (64.3% [95% CI: 62.2% to 66.3%]) experienced presenteeism.

Table 1 presents the descriptive statistics of the respondents' demographics. A total of 2,064 women completed the dysmenorrhea questionnaire. The severe group, based on the dysmenorrhea score, consisted of 902 participants (43.7%), excluding those with missing scores. Younger participants tended to have a higher prevalence in the severe group. Approximately 70% of participants were office workers. Based on the dysmenorrhea score, the proportion of participants with asthma and allergies was significantly higher in the severe group than in the normal group.

### Degree of impairment of work productivity and HRQoL

Table 2 shows the median values of each WPAI-GH indicator and summary scores of SF-36v2. Absenteeism, presenteeism, overall work impairment, and activity impairment in the WPAI-GH group were significantly higher than those in the severe group (Fig 1A-1D). Regarding HRQoL, the scores for all three components of the SF-36v2, namely the PCS, MCS, and RCS, were significantly lower in the severe group than in the normal group (Fig 1E-1G).

### Multivariable analysis

In the multiple regression analysis, even after adjusting cofounding factors, the severity of the dysmenorrhea score was significant factor of impaired absenteeism (beta [b]=2.14 [95% CI: 0.66 to 3.62]), presenteeism (b = 9.07 [95% CI: 6.92 to 11.22]), overall work impairment (b = 9.52 [95%CI: 7.24–11.80]) and activity impairment (b = 10.47 [95% CI: 7.24 to 11.80]) (Fig 2A-2D). In addition, the absence of comorbidities (b = −3.04 [95%CI: −5.21 to −0.88]) was positively associated with absenteeism. Comorbidity with heart disease was slightly associated with greater impairment in presenteeism, overall work impairment, and activity impairment compared to no heart disease.

Interestingly, low annual household income was associated with impaired absenteeism (b = 3.71 [95% CI: 1.29 to 6.13]), presenteeism (b = 3.51 [95% CI: 0.10 to 7.12]), overall work impairment (b = 4.42 [95% CI: 0.70 to 8.14]) and activity impairment (b = 3.00 [95% CI: 0.70 to 8.14]). Living in urban areas was significantly associated with greater impairment in presenteeism (b = 3.34 [95% CI: 1.16 to 5.52]), overall work impairment (b = 3.33 [95% CI: 1.02 to 5.64]), and activity impairment (b = 2.85 [95% CI: 1.02 to 5.64]) compared to living in rural areas.

The severe dysmenorrhea score was significantly associated with reduced PCS (b = −2.76 [95% CI: −3.42 to −2.09]), MCS (b = −3.96 [95% CI: −4.77 to −3.16]), and RCS (b = −1.08 [95% CI: −1.98 to −0.17]). Absence of comorbidities was positively associated with higher physical QoL (b = 2.05 [95% CI: 1.07 to 3.02]) and higher mental QoL (b = 1.24 [95% CI: 0.06 to 2.42]). Asthma was associated with lower PCS scores, whereas heart disease was associated with lower MCS scores. Interestingly, low income was a significant factor for reduced MCS (b = −2.66 [95% CI: −3.99 to −1.33]) and RCS (b = −1.89 [95% CI: −3.38 to −0.40]) (Fig 2E-2G).

## Discussion

### Prevalence of dysmenorrhea

In this study, a dysmenorrhea score of ≥3 was defined as indicating severe dysmenorrhea. Under this definition, 43.7% of the participants had severe dysmenorrhea. In a web-based cross-sectional study conducted in France, 66% of women

**Table 1. Participant characteristics in the overall cohort and in groups classified by dysmenorrhea score.**

| | Overall | Dysmenorrhea score | | |
| --- | --- | --- | --- | --- |
| Characteristic | N = 2,555 | Normal N = 1,162 | Severe N = 902 | p-value[2] |
| **Age, n (%)** | | | | **0.005** |
| **20-29** | 714 (28%) | 277 (24%) | 263 (29%) | |
| **30-39** | 971 (38%) | 446 (38%) | 352 (39%) | |
| **40-49** | 870 (34%) | 439 (38%) | 287 (32%) | |
| **Body mass index[1]** | 21.48 (19.56, 24.21) | 21.51 (19.57, 24.14) | 21.64 (19.84, 25.10) | 0.017 |
| **Gravidity[1]** | 1.00 (1.00, 3.00) | 1.00 (1.00, 3.00) | 1.00 (1.00, 3.00) | **<0.001** |
| **Parity[1]** | 1.00 (1.00, 2.00) | 1.00 (1.00, 2.00) | 1.00 (1.00, 2.00) | **<0.001** |
| **Area of residence, n (%)** | | | | 0.852 |
| **Urban** | 1,497 (59%) | 695 (60%) | 532 (59%) | |
| **Rural** | 1,015 (40%) | 462 (40%) | 367 (41%) | |
| **Overseas** | 8 (0.3%) | 5 (0.4%) | 3 (0.3%) | |
| **Employment, n (%)** | | | | **0.049** |
| **Office worker** | 1,785 (70%) | 790 (68%) | 645 (72%) | |
| **Part-timer** | 631 (25%) | 314 (27%) | 199 (22%) | |
| **Freelancer** | 88 (3.4%) | 40 (3.4%) | 35 (3.9%) | |
| **Business owner** | 14 (0.5%) | 7 (0.6%) | 6 (0.7%) | |
| **Others** | 37 (1.4%) | 11 (0.9%) | 17 (1.9%) | |
| **Highest level of education, n (%)** | | | | 0.819 |
| **Higher education** | 1,062 (42%) | 495 (43%) | 391 (43%) | |
| **Post secondary education** | 754 (30%) | 350 (30%) | 256 (28%) | |
| **Highschool** | 640 (25%) | 289 (25%) | 230 (25%) | |
| **Junior high** | 64 (2.5%) | 28 (2.4%) | 25 (2.8%) | |
| **Type of industry, n (%)** | | | | 0.813 |
| **Construction** | 73 (2.9%) | 32 (2.8%) | 26 (2.9%) | |
| **Manufacturing** | 202 (7.9%) | 89 (7.7%) | 72 (8.0%) | |
| **Electricity, Gas, Heat, and Water Supply** | 20 (0.8%) | 7 (0.6%) | 11 (1.2%) | |
| **Information and Communications** | 88 (3.4%) | 38 (3.3%) | 26 (2.9%) | |
| **Transport and Postal Services** | 66 (2.6%) | 34 (2.9%) | 25 (2.8%) | |
| **Wholesale and Retail Trade** | 226 (8.8%) | 99 (8.5%) | 82 (9.1%) | |
| **Finance and Insurance** | 94 (3.7%) | 44 (3.8%) | 35 (3.9%) | |
| **Real Estate** | 23 (0.9%) | 7 (0.6%) | 11 (1.2%) | |
| **Accommodation and Food Services** | 131 (5.1%) | 66 (5.7%) | 43 (4.8%) | |
| **Medical and Welfare Services** | 812 (32%) | 373 (32%) | 280 (31%) | |
| **Education** | 275 (11%) | 136 (12%) | 93 (10%) | |
| **Other Services** | 308 (12%) | 132 (11%) | 114 (13%) | |
| **Others** | 237 (9.3%) | 105 (9.0%) | 84 (9.3%) | |
| **Annual household income, n (%)** | | | | 0.414 |
| **High** | 278 (11%) | 136 (12%) | 99 (11%) | |
| **Middle** | 998 (39%) | 492 (42%) | 362 (40%) | |
| **Low** | 1,183 (46%) | 534 (46%) | 441 (49%) | |
| **Comorbidities** | | | | |
| **Without, n (%)** | 1,175 (46.1%) | 556 (47.8%) | 386 (42.8%) | **0.022** |
| **HT, n (%)** | 101 (4.0%) | 46 (4.0%) | 40 (4.4%) | 0.591 |
| **DL, n (%)** | 111 (4.4%) | 54 (4.6%) | 42 (4.7%) | 0.992 |

*(Continued)*

**Table 1.** (Continued)

|  | Overall | Dysmenorrhea score |  |  |
|---|---|---|---|---|
| DM, n (%) | 36 (1.4%) | 14 (1.2%) | 14 (1.6%) | 0.499 |
| Asthma, n (%) | 313 (12.3%) | 130 (11.2%) | 128 (14.2%) | **0.041** |
| Allergy, n (%) | 551 (21.6%) | 228 (19.6%) | 225 (24.9%) | **0.004** |
| Liver disease, n (%) | 14 (0.5%) | 6 (0.5%) | 7 (0.8%) | 0.459 |
| Heart disease, n (%) | 40 (1.6%) | 20 (1.7%) | 13 (1.4%) | 0.615 |
| Tuberculosis, n (%) | 3 (0.1%) | 1 (0.1%) | 2 (0.2%) | 0.584 |
| Autoimmune disease, n (%) | 37 (1.5%) | 24 (2.1%) | 8 (0.9%) | **0.032** |
| Anemia, n (%) | 553 (21.7%) | 242 (20.8%) | 206 (22.8%) | 0.271 |
| Stroke, n (%) | 5 (0.2%) | 2 (0.2%) | 1 (0.1%) | >0.999 |
| Cancer, n (%) | 33 (1.3%) | 10 (0.9%) | 17 (1.9%) | **0.042** |
| Other comorbidities, n (%) | 302 (11.8%) | 136 (11.7%) | 116 (12.9%) | 0.426 |

[1]Median (Q1, Q3).

[2]Chi-squared test and Wilcoxon rank sum test. t-test for BMI.

Abbreviations: HT, hypertension; DL, dyslipidemia; DM, diabetes mellitus.

**Table 2. Comparison of WPAI-GH and SF-36v2 scores between normal and severe groups classified by dysmenorrhea score.**

|  | Dysmenorrhea score |  |  |
|---|---|---|---|
| **Summary scores** | **Normal** N = 1,162[1] | **Severe** N = 902[1] | **p-value[2]** |
| **WPAI-GH** |  |  |  |
| **Absenteeism, %** | 0 (0, 0) | 0 (0, 0) | **<0.001** |
| **Presenteeism, %** | 10 (0, 30) | 20 (0, 50) | **<0.001** |
| **Overall work impairment, %** | 10 (0, 30) | 26 (0, 50) | **<0.001** |
| **Activity impairment, %** | 10 (0, 30) | 30 (10, 50) | **<0.001** |
| **SF-36v2[3]** |  |  |  |
| **Physical functioning (PF)** | 54.4 (51.7, 57.1) | 54.4 (48.9, 57.1) | **<0.001** |
| **Role physical (RP)** | 54 (45, 57) | 48 (39, 57) | **<0.001** |
| **Bodily pain (BP)** | 45 (39, 54) | 39 (34, 44) | **<0.001** |
| **General health (GH)** | 55 (47, 60) | 49 (42, 57) | **<0.001** |
| **Vitality (VT)** | 47 (41, 53) | 44 (35, 50) | **<0.001** |
| **Social functioning (SF)** | 52 (41, 58) | 46 (41, 58) | **<0.001** |
| **Role emotional (RE)** | 49 (42, 57) | 46 (35, 53) | **<0.001** |
| **Menal health (MH)** | 50 (42, 57) | 47 (39, 55) | **<0.001** |
| **Physical component summary (PCS)** | 53 (49, 58) | 51 (46, 56) | **<0.001** |
| **Mental component summary (MCS)** | 49 (42, 54) | 44 (37, 51) | **<0.001** |
| **Role/Social component summary (RCS)** | 49 (42, 55) | 47 (39, 55) | **0.015** |

[1]Median (Q1, Q3)

[2]Wilcoxon rank sum test

[3]Norm-based score

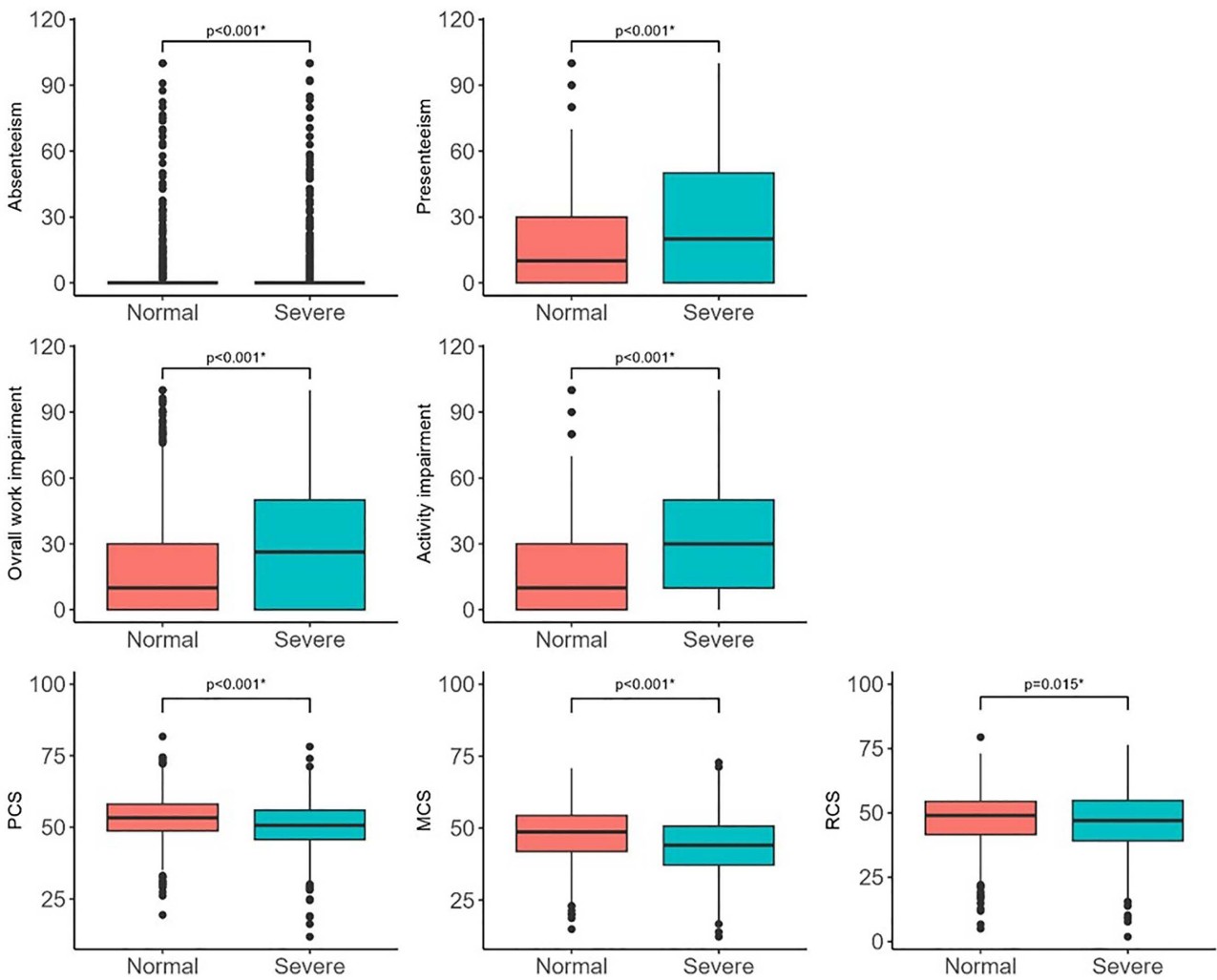

**Fig 1. Comparison of work productivity and QoL between women with normal and severe dysmenorrhea scores.** A, absenteeism; B, presenteeism; C, overall work impairment; D, activity impairment; E, physical component summary (PCS); F, mental component summary (MCS); G, role/social component summary (RCS) scores. Statistical significance was determined using Wilcoxon rank-sum test. Asterisks (*) indicate statistically significant differences (p<0.05).

aged ≤24 years, 57% of women aged 25–34 years, and 50% of women aged 35–44 years reported experiencing dysmenorrhea as defined by the VAS [8]. The results of this study are consistent with the prevalence reported in previous studies, suggesting that the dysmenorrhea score is a simple and useful tool for assessing pain severity and defining dysmenorrhea for statistical evaluation.

## Prevalence of absenteeism and presenteeism

In this study cohort, 17.5% of the participants experienced absenteeism and 64.3% experienced presenteeism due to health issues. Notably, the specific diseases or severities of menstrual-related symptoms that caused absenteeism or presenteeism were not clear in this study. A nationwide Dutch survey showed that 13.8% of women reported absenteeism and 80.7% reported presenteeism during menstruation [5]. In a Brazilian web-based study, 44.2% of women reported

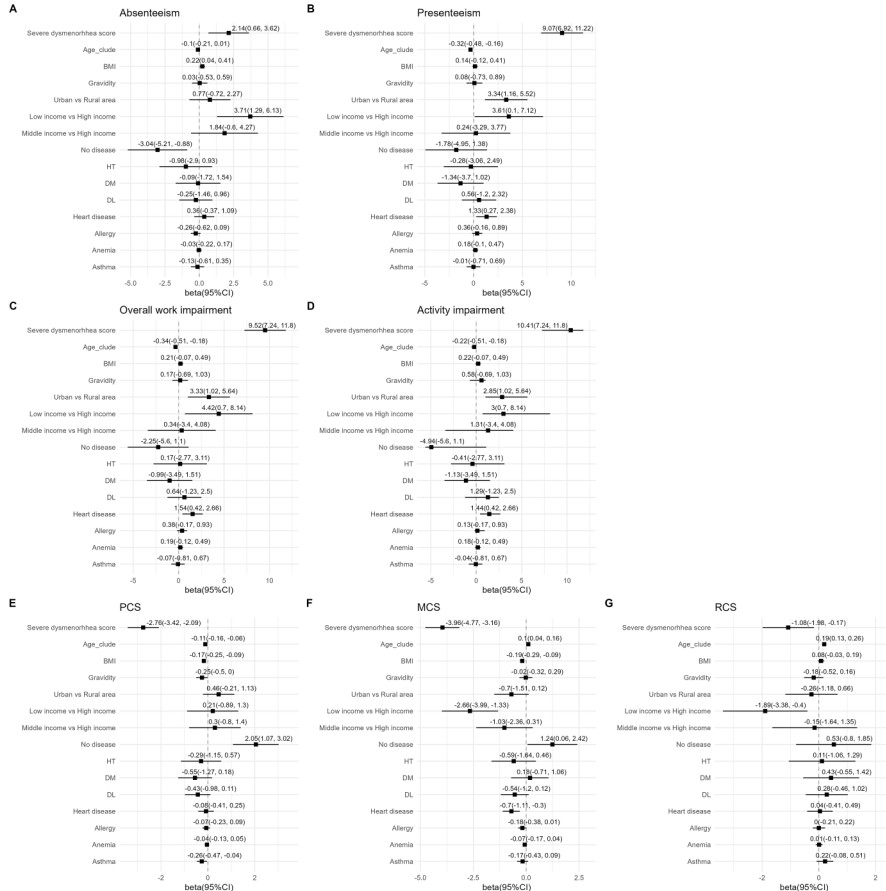

**Fig 2. Multiple regression analyses of factors associated with work productivity and QoL.** Forest plots show point estimates of beta coefficients and 95% confidence intervals (CIs) for each variable in the multiple regression models. A, absenteeism; B, presenteeism; C, overall work impairment; D, activity impairment; E, physical component summary (PCS); F, mental component summary (MCS); G, role/social component summary (RCS) scores.

presenteeism, and women with severe menstrual flow were more likely to report presenteeism [26]. The prevalence of absenteeism and presenteeism observed in this study was generally consistent with previous reports.

## Dysmenorrhea and work productivity

In this study, we demonstrated that the group with severe dysmenorrhea experienced a significant decrease in work productivity. Even after adjusting for other factors, the impact of dysmenorrhea score on work productivity remained significant. Additionally, previous studies have demonstrated that dysmenorrhea affects work productivity. Schoep et al. showed that higher VAS scores were significantly associated with higher levels of absenteeism and presenteeism [5]. Similarly, Cook and van den Hoek demonstrated in their survey that presenteeism was positively correlated with dysmenorrhea severity [6]. The findings of this and previous studies suggest that addressing pain symptoms is important for improving work productivity. In a prospective study conducted in Japan, the average overall work impairment among patients with dysmenorrhea was 50.2%, which decreased to 31.7% 60 days after the introduction of low-dose estrogen-progestin [7]. In this study, the dysmenorrhea score included the status of analgesic use. Therefore, in the group with severe scores, it is likely that the patients were already using sufficient analgesics. For these patients, additional treatment, including low-dose estrogen-progestin may improve work productivity.

## Dysmenorrhea and HRQoL

In this study, severe dysmenorrhea was associated with decreased QoL across all physical, mental, and psychosocial indicators. Previous studies have investigated the impact of dysmenorrhea on QoL. Fernandez et al. reported that women with dysmenorrhea have significantly lower physical and mental QoL scores [8]. Similarly, Mizuta et al. demonstrated that the severity of dysmenorrhea is associated with reduced physical and environmental QoL [9]. Shimamoto et al. demonstrated that lower abdominal pain was a significant factor in reducing HRQoL [10]. Dysmenorrhea should be noted not only for its physical effects but also for its potential to decrease both mental and social QoL.

## Validity of diagnosis of dysmenorrhea

In this study, 43.7% of participants were in the severe group. This classification aligns well with prevalence rates reported in previous studies, indicating that this scoring method neither significantly underestimated nor overestimated the prevalence of dysmenorrhea. Therefore, the dysmenorrhea score is a promising practical screening tool for the diagnosis of dysmenorrhea.

While many previous studies have used VAS or NRS as pain indicators [1], these measures have limitations, particularly in capturing pain severity that patients may not consciously perceive or pain masked by analgesic use. The Menstrual Distress Questionnaire [27] provides a detailed assessment of various menstrual-related symptoms, including premenstrual syndrome. However, it has the drawback of being complex owing to many questionnaire items. The WaLIDD score [28] is especially effective for screening dysmenorrhea, which affects work ability and QoL. However, it consists of four questions and depends on self-reported pain intensity, which may lead to underestimation when analgesics are used. In contrast, the dysmenorrhea score has the advantage of being a simple two-question tool that can assess pain severity considering analgesic use.

In clinical trials, the dysmenorrhea score has been used to evaluate treatment efficacy in patients with endometriosis [12,13]. Although previous studies have attempted to diagnose dysmenorrhea using complex questionnaires alone [29,30], a simpler indicator is preferable for screening untreated populations for potential endometriosis. In this context, the dysmenorrhea score may serve as a valuable screening measure. However, it should be noted that the accuracy of using a dysmenorrhea score of ≥3 as a screening tool for patients needing treatment has not been fully studied. Although our exploratory analysis also showed score 3 as a reasonable inflection point (S1 Fig), it would be necessary to conduct prospective clinical studies to demonstrate its validity.

## Socioeconomic factors

Interestingly, low annual household income was associated with higher levels of absenteeism and presenteeism. Previous studies have shown that higher income is associated with lower levels of absenteeism [31]. It is hypothesized that individuals with greater financial resources are less likely to be absent from work when faced with adverse circumstances [32].

In line with this, previous research has indicated that higher income levels are linked to increased presenteeism [31]. This result differs from our current finding that a lower income is associated with a higher level of presenteeism. A possible hypothesis is that individuals with lower incomes frequently experience a stronger financial imperative to work even when they have severe dysmenorrhea. Consequently, these workers are more likely to engage in presenteeism—that is, continuing to work despite illness. In contrast, those with higher incomes generally possess greater flexibility and resources, enabling them to take leave when sick and effectively manage both their workload and financial stability. Furthermore, the relationship between financial status and absenteeism and presenteeism may differ depending on the nature of the disease. In a study of a different condition, it was shown that low-income workers in Japan exhibited increased presenteeism during the COVID-19 pandemic, as financial hardship compelled them to continue working even when feeling unwell [33]. To explore this hypothesis, qualitative research methods, such as interview surveys, would be valuable for examining

differences by disease and severity. In particular, detailed interviews could elucidate when individuals choose to take sick leave, how their symptoms influence work productivity, what strategies are adopted to maintain productivity, and the associated costs of these adaptations.

Additionally, living in an urban area was significantly associated with greater impairment in presenteeism and activity than living in a rural area. Although in patients with axial spondyloarthritis, it has been found that those living in rural areas reported higher levels of presenteeism in the United Kingdom [34]. In contrast, in the context of diabetes, the levels of both absenteeism and presenteeism are higher in urban areas than in rural areas [35]. Regarding dysmenorrhea, the differences between urban and non-urban areas may be related to factors such as health literacy, knowledge, and awareness regarding dysmenorrhea and its management; lifestyle factors such as physical activity; and differences in the working population. The impact of disease on absenteeism and presenteeism in urban and rural areas may differ depending on disease characteristics. However, few studies have analyzed regional differences in the impact of dysmenorrhea on absenteeism and presenteeism, and further research is warranted.

## Limitations

This study had several limitations. First, as this was a cross-sectional study, causality could not be established. Therefore, whether improvements in dysmenorrhea would truly contribute to improved work productivity or QoL remains unclear. Second, because the responses were based on patient recall, recall bias may have occurred; that is, individuals with more severe symptoms may be more likely to recall their degree of work impairment. This could result in those able to recall more conditions also reporting higher dysmenorrhea scores and greater impairment in QoL and work productivity, leading to a possible spurious correlation. Furthermore, since comorbidities were self-reported by the patients, the accuracy of these responses may be limited, and the severity or control status of the comorbidities could not be determined. For example, severe dysmenorrhea may be accompanied by severe anemia, which itself can negatively affect QoL and work productivity; if anemia is underreported, residual confounding may remain even after multivariate analysis. In addition, since recruitment for research participation was conducted through voluntary enrollment via smartphone app, there is a potential for selection bias among the participants. Specifically, individuals with higher health awareness and health literacy may be overrepresented. Therefore, caution is needed when generalizing the findings of this study to all Japanese women, and even more so when generalizing to women in other countries.

## Conclusions

Severe dysmenorrhea, defined as a dysmenorrhea score of ≥3, was associated with impaired work productivity and reduced QoL. QoL can be affected not only physically, but also mentally or socially by dysmenorrhea. Screening at a threshold of three points for the dysmenorrhea score and providing hormone therapy and/or workplace adjustments to those who meet this criterion may contribute to improvements in work productivity and QoL.

## Supporting information

**S1 Fig. Relationship between dysmenorrhea score and presenteeism, activity impairment, physical component summary (PCS), mental component summary (MCS), and role/social component summary (RCS) scores.** The blue curve represents a quartic polynomial fit, and the red dashed lines indicate the positions of the inflection points. For absenteeism, no clear inflection point was found and thus omitted.
(JPG)

## Acknowledgments

We would like to thank Editage (www.editage.com) for English language editing.

## Author contributions

**Conceptualization:** Osamu Wada-Hiraike.

**Data curation:** Maika Nariai, Eri Maeda, Masayo Matsuzaki.

**Formal analysis:** Maika Nariai, Osamu Wada-Hiraike, Eri Maeda, Masayo Matsuzaki, Maho Furukawa, Yuki Enomoto, Hiromi Ga, Risa Takai.

**Funding acquisition:** Osamu Wada-Hiraike, Yutaka Osuga.

**Investigation:** Maika Nariai.

**Methodology:** Osamu Wada-Hiraike.

**Project administration:** Osamu Wada-Hiraike.

**Supervision:** Osamu Wada-Hiraike, Eri Maeda, Masayo Matsuzaki, Mayuyo Mori-Uchino, Miyuki Harada, Yutaka Osuga, Yasushi Hirota.

**Validation:** Osamu Wada-Hiraike, Eri Maeda, Masayo Matsuzaki, Mayuyo Mori-Uchino, Maho Furukawa, Yuki Enomoto, Hiromi Ga, Risa Takai, Miyuki Harada, Yasushi Hirota.

**Visualization:** Osamu Wada-Hiraike.

**Writing – original draft:** Maika Nariai, Osamu Wada-Hiraike.

**Writing – review & editing:** Maika Nariai, Osamu Wada-Hiraike.

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
