## [Decision Letter · Decision Letter 0]

27 Aug 2025

Dear Dr. Wada-Hiraike,

Thank you for submitting your manuscript to PLOS ONE. After careful consideration, we feel that it has merit but does not fully meet PLOS ONE’s publication criteria as it currently stands. Therefore, we invite you to submit a revised version of the manuscript that addresses the points raised during the review process.

We look forward to receiving your revised manuscript.

Kind regards,

Guilherme Tavares de Arruda

Academic Editor

PLOS ONE

Journal Requirements: 

4. Please amend your authorship list in your manuscript file to include author Miyuki Harada.

Additional Editor Comments:

Congratulations on this relevant work! I advise you to revise your manuscript according to the reviewers' comments.

Reviewers' comments:

Reviewer's Responses to Questions

**Comments to the Author**

1. Is the manuscript technically sound, and do the data support the conclusions?

Reviewer #1: Yes

Reviewer #2: Yes

2. Has the statistical analysis been performed appropriately and rigorously?

Reviewer #1: Yes

Reviewer #2: Yes

3. Have the authors made all data underlying the findings in their manuscript fully available?

Reviewer #1: Yes

Reviewer #2: Yes

4. Is the manuscript presented in an intelligible fashion and written in standard English?

Reviewer #1: Yes

Reviewer #2: Yes

Reviewer #1: This is a well-designed and relevant study that has significant public health implications. Dysmenorrhoea is the commonest clause of work absenteeism and this work has thrown more light on the objective management of the condition. After addressing some minor typographical and presentation issues, the manuscript would be suitable for publication.

Reviewer #2: To the Editors of PLOS ONE,

Thank you for the opportunity to review the manuscript PONE-D-25-37160, "Effects of dysmenorrhea on work productivity and quality of life in Japanese women: A large-scale web-based cross-sectional study."

This study presents a large-scale cross-sectional analysis of the impact of dysmenorrhea on work productivity and quality of life (QoL) in Japanese women, utilizing data from a web-based questionnaire. The authors' use of a dysmenorrhea score, which accounts for menstrual pain severity, its impact on daily activities, and medication use, is a particularly interesting and relevant approach. The findings that severe dysmenorrhea is a significant predictor of impaired work productivity and reduced physical, mental, and social QoL are important contributions to the literature. The manuscript is well-written, and the statistical analysis is appropriate given the study design.

While the study is robust in its methodology and contributes valuable information, I have several major concerns and questions that need to be addressed before it can be considered for publication. My recommendation is to accept after minor revisions, provided the authors can adequately respond to these points.

Comments and Questions for the Authors:

1. Causality and Clinical Implications: The study's cross-sectional design is a significant limitation, as acknowledged by the authors4. However, the conclusion that "the application of a dysmenorrhea score may facilitate the screening of dysmenorrhea in clinical practice" and that a score of

≥3 "appears to be a reasonable and valid criterion" for treatment eligibility is a very strong statement. Given the lack of causal inference, how can the authors definitively conclude that a score of ≥3 is a valid treatment threshold, or that therapeutic interventions (e.g., hormone therapy or workplace adjustments) would lead to the observed improvements?

Please elaborate on how these conclusions can be supported without prospective data and discuss how future research could address this limitation.

2. Selection Bias and Generalizability: The study population was recruited through a smartphone application, Luna Luna. This sampling method may introduce selection bias, as women who use such applications may have different demographics, health-seeking behaviors, and symptom awareness compared to the general Japanese female population. Please provide a more detailed discussion on the potential for selection bias and its impact on the generalizability of the findings to all Japanese women.

3. Self-Reported Data and Recall Bias: The study relies on self-reported data, including the dysmenorrhea score, comorbidities, and work impairment. The authors themselves note the possibility of recall bias, particularly among those with more severe symptoms. Furthermore, comorbidities were self-reported, and their severity or control status was not assessed, which may have biased the outcomes. Please provide a more in-depth discussion on how these limitations might have affected the results and what steps, if any, were taken to mitigate them.

4. Socioeconomic Factors: The finding that low annual household income was associated with higher levels of presenteeism is intriguing and seemingly contradictory to some previous studies. The authors' hypothesis that financial necessity may compel women with lower incomes to work while ill is plausible. Please expand this section of the discussion, providing a more detailed theoretical framework for this finding and suggestions for how this relationship could be further investigated in future studies.

5. Dysmenorrhea Score Validation: One of the study's aims is to demonstrate the validity of the dysmenorrhea score as a screening tool. While the authors compare their prevalence rates with other studies using different scales, a more robust validation would strengthen the manuscript. Were any analyses conducted to determine the sensitivity and specificity of the ≥3 cutoff? Providing a more thorough justification for this specific threshold would be highly valuable.

I commend the authors on their diligent work and look forward to their responses. The manuscript is a valuable contribution to the field, and addressing these points will significantly improve its quality and impact.

Sincerely,

Eduardo Batista Cândido M.D.; PhD

**Do you want your identity to be public for this peer review?** For information about this choice, including consent withdrawal, please see our Privacy Policy

Reviewer #1: No

Reviewer #2: No

---

## [Author Response · Author response to Decision Letter 1]

27 Oct 2025

Manuscript ID: PONE-D-25-37160

Dear Editor and Reviewers,

We thank you for the thoughtful and constructive comments on our manuscript. We have carefully addressed each point raised and believe these revisions have substantially improved the quality of our work. Below is our point-by-point response to each comment.

Journal Requirements:

Requirement 1: Please ensure that your manuscript meets PLOS ONE's style requirements.

Response: We have reformatted the entire manuscript according to PLOS ONE style templates. [Changes made in: entire manuscript]

Requirement 2: Grant information discrepancy between Funding Information and Financial Disclosure sections.

Response: We have corrected the funding information to ensure consistency between both sections. The correct grant numbers are now accurately listed in the Funding Information section. [Changes made in: Financial Disclosure and Financial Disclosure]

Requirement 3: Please provide a complete Data Availability Statement.

Response: We have added the following Data Availability Statement: " The data used in this study are not publicly available due to privacy or policy restrictions. However, summarized or anonymized data may be provided by the authors upon reasonable request and subject to approval by the corresponding author." [Changes made in: Data Availability section]

Requirement 4: Please amend authorship list to include Miyuki Harada.

Response: We have added Miyuki Harada to the authorship list with appropriate affiliation information.

Requirement 5: Reference list review.

Response: We have thoroughly reviewed and updated all references. No retracted articles were cited in our manuscript.

Reviewer #1 Comments:

Comment: This is a well-designed and relevant study with significant public health implications. After addressing minor typographical and presentation issues, the manuscript would be suitable for publication.

Response: Thank you for your positive evaluation. We have carefully reviewed the manuscript for typographical and presentation issues and made necessary corrections throughout. [Changes made in: throughout manuscript]

Reviewer #2 Comments:

Comment 1: Causality and Clinical Implications: The study's cross-sectional design is a significant limitation, as acknowledged by the authors. However, the conclusion that "the application of a dysmenorrhea score may facilitate the screening of dysmenorrhea in clinical practice" and that a score of ≥3 "appears to be a reasonable and valid criterion" for treatment eligibility is a very strong statement. Given the lack of causal inference, how can the authors definitively conclude that a score of ≥3 is a valid treatment threshold, or that therapeutic interventions (e.g., hormone therapy or workplace adjustments) would lead to the observed improvements?

Please elaborate on how these conclusions can be supported without prospective data and discuss how future research could address this limitation.

Response: We acknowledge this important limitation. We have revised our conclusions to be more cautious and appropriate for cross-sectional data. We have added extensive discussion of the need for prospective validation studies. [Changes made in: Abstract lines 28-29, Discussion lines 304-308]

Comment 2: Selection Bias and Generalizability: The study population was recruited through a smartphone application, Luna Luna. This sampling method may introduce selection bias, as women who use such applications may have different demographics, health-seeking behaviors, and symptom awareness compared to the general Japanese female population. Please provide a more detailed discussion on the potential for selection bias and its impact on the generalizability of the findings to all Japanese women.

Response: Thank you for this crucial point. In a study targeting users of the same application, 310,000 users recorded at least ten menstrual cycles over the two-year period from 2016 to 2017 (Obstet Gynecol. 2020;136(4):666-674), suggesting that the app has a sufficiently large number of active users. However, as the reviewer pointed out, there is considerable selection bias since the participants are limited to menstrual tracking app users, and also a significant selection bias due to voluntary participation in the questionnaire survey. We have expanded our discussion of selection bias and generalizability limitations by acknowledging limitations in health-seeking behaviors and symptom awareness. [Changes made in: Discussion section, lines 352-364]

Comment 3: Self-Reported Data and Recall Bias: The study relies on self-reported data, including the dysmenorrhea score, comorbidities, and work impairment. The authors themselves note the possibility of recall bias, particularly among those with more severe symptoms. Furthermore, comorbidities were self-reported, and their severity or control status was not assessed, which may have biased the outcomes. Please provide a more in-depth discussion on how these limitations might have affected the results and what steps, if any, were taken to mitigate them.

Response: Thank you again for this critical review. We agree with the reviewer’s comments, and we would be grateful if you could refer to lines 349-351 of the Discussion section, where we have already addressed the issue of recall bias. Additionally, in the questionnaire, instead of allowing open-ended responses, we asked participants to “select all diseases you have been diagnosed with from the following list.” This approach was designed to encourage answers that focus on actual diagnoses. However, individuals with severe dysmenorrhea symptoms may be more sensitive to changes in their health status and therefore more likely to recall and report a greater number of diagnosed conditions, resulting in recall bias. As a result, for example, those who can recall more diseases (i.e., have more comorbidities) may report higher dysmenorrhea scores and appear to have greater impairment in QoL and work productivity, leading to a possible spurious correlation.

Moreover, limited data on the severity of the diseases presents another issue; for instance, those with severe dysmenorrhea may have severe anemia, and anemia could contribute to reduced QoL or work productivity. If the presence of anemia is underreported, it may not be fully accounted for as a confounder in multivariate analysis. We discussed these issues in greater detail in the manuscript. [Changes made in: Discussion section, lines 352-364]

Comment 4: Socioeconomic Factors: The finding that low annual household income was associated with higher levels of presenteeism is intriguing and seemingly contradictory to some previous studies. The authors' hypothesis that financial necessity may compel women with lower incomes to work while ill is plausible. Please expand this section of the discussion, providing a more detailed theoretical framework for this finding and suggestions for how this relationship could be further investigated in future studies.

Response: We have provided detailed theoretical framework explaining financial necessity hypothesis and future research suggestions. We have also added citations of previous studies supporting this hypothesis to the reference list. [Changes made in: Discussion section, lines 318-332]

Comment 5: 5. Dysmenorrhea Score Validation: One of the study's aims is to demonstrate the validity of the dysmenorrhea score as a screening tool. While the authors compare their prevalence rates with other studies using different scales, a more robust validation would strengthen the manuscript. Were any analyses conducted to determine the sensitivity and specificity of the ≥3 cutoff? Providing a more thorough justification for this specific threshold would be highly valuable.

Response: We acknowledge this limitation and consider it an important topic for future research. To date, no validation studies have been conducted on the diagnostic accuracy of the dysmenorrhea score. However, many clinical studies have consistently and widely applied a cutoff of 3 or higher to define severe cases and treatment eligibility (Fertil Steril. 2008;90: 1583-1588, Fertil Steril. 2016;106: 1807-1814, Fertil Steril. 2020;113: 167-175). Therefore, in the present study, we also adopted a threshold of 3. As an exploratory analysis, we examined the relationship between the dysmenorrhea score and outcome measures (the four measures of the WPAI-GH and the SF-36v2 summary scores). Since these outcomes showed a monotonic increase or decrease with the dysmenorrhea score, we considered that the point at which the change in the outcome variable is greatest (the inflection point) could serve as a potential threshold for the dysmenorrhea score. Accordingly, we fitted a quartic curve and plotted the inflection points as described below (Suppl Figure 1). For absenteeism, no clear inflection point was found, and for overall work impairment, the results were similar to those for presenteeism and thus omitted. As a result, the inflection points were located between 1.6 and 2.5. Based on this, setting the threshold at 3 for the dysmenorrhea score appears to be generally appropriate. In other words, the difference in outcome measures is substantial between a dysmenorrhea score of 2 or lower and 3 or higher. However, it should be noted that this threshold is derived from outcome metrics, not from validation studies of diagnostic accuracy.

Furthermore, for the validation of diagnostic accuracy of dysmenorrhea score, new challenges will emerge: There is currently no standardized definition of "dysmenorrhea"; different studies use varying definitions. Therefore, it is necessary to clarify what the sensitivity and specificity would be measured against—for example, whether the diagnostic accuracy is for self-reported severity or for diagnoses based on other criteria such as the VAS score. In the future, we plan to address these issues and conduct a sensitivity and specificity analysis of the dysmenorrhea score. We may be able to present these findings in future publications.

S1 Fig. Relationship between dysmenorrhea score and presenteeism, activity impairment, physical component summary (PCS), mental component summary (MCS), and role/social component summary (RCS) scores. The blue curve represents a quartic polynomial fit, and the red dashed lines indicate the positions of the inflection points.

---

## [Editor Report · Decision Letter 1]

10 Nov 2025

Effects of dysmenorrhea on work productivity and quality of life in Japanese women: A large-scale web-based cross-sectional study

PONE-D-25-37160R1

Dear Dr. Wada-Hiraike,

We’re pleased to inform you that your manuscript has been judged scientifically suitable for publication and will be formally accepted for publication once it meets all outstanding technical requirements.

Kind regards,

Guilherme Tavares de Arruda

Academic Editor

PLOS ONE
---

## [Editor Report · Acceptance letter]

PONE-D-25-37160R1

PLOS ONE

Dear Dr. Wada-Hiraike,

I'm pleased to inform you that your manuscript has been deemed suitable for publication in PLOS ONE. Congratulations! Your manuscript is now being handed over to our production team.

Kind regards,

on behalf of

Dr. Guilherme Tavares de Arruda

Academic Editor

PLOS ONE